# Uncertainty Calibration for Ensemble-Based Debiasing Methods

**Ruibin Xiong**[1,2,3*], **Yimeng Chen**[2,4*†], **Liang Pang**[2,5], **Xueqi Cheng**[1,2],
**Zhiming Ma**[2,4] and **Yanyan Lan**[6‡]

[1]CAS Key Laboratory of Network Data Science and Technology,
Institute of Computing Technology, Chinese Academy of Sciences
[2]University of Chinese Academy of Sciences [3]Baidu Inc.
[4]Academy of Mathematics and Systems Science, Chinese Academy of Sciences
[5]Data Intelligence System Research Center,
Institute of Computing Technology, Chinese Academy of Sciences
[6] Institute for AI Industry Research, Tsinghua University
{xiongruibin18, chenyimeng14}@mails.ucas.ac.cn,
{cxq, pangliang}@ict.ac.cn,
mazm@amt.ac.cn,lanyanyan@tsinghua.edu.cn

## Abstract

Ensemble-based debiasing methods have been shown effective in mitigating the reliance of classifiers on specific dataset bias, by exploiting the output of a bias-only model to adjust the learning target. In this paper, we focus on the bias-only model in these ensemble-based methods, which plays an important role but has not gained much attention in the existing literature. Theoretically, we prove that the debiasing performance can be damaged by inaccurate uncertainty estimations of the bias-only model. Empirically, we show that existing bias-only models fall short in producing accurate uncertainty estimations. Motivated by these findings, we propose to conduct calibration on the bias-only model, thus achieving a three-stage ensemble-based debiasing framework, including bias modeling, model calibrating, and debiasing. Experimental results on NLI and fact verification tasks show that our proposed three-stage debiasing framework consistently outperforms the traditional two-stage one in out-of-distribution accuracy.

## 1 Introduction

Machine learning models have achieved remarkable performance on natural language understanding [10; 7; 28] and computer vision [16; 17]. However, observations have shown that these models have difficulties in generalizing well in out-of-distribution settings [30; 40; 2; 11], which limits their applications to real-world scenarios. A major cause of this failure is the reliance of the model on specific *dataset bias* [33]. For instance, McCoy et al. [23] have shown that sentence pairs with high word overlaps in MNLI are easy to be classified as the label 'entailment', even if they have different relations.

A growing body of literature recognizes debiasing as an important direction in machine learning and natural language processing [38; 3; 4; 34]. Within these works, *ensemble-based debiasing* (EBD) methods [15; 22; 8; 39; 5] have caused considerable interest within the community, as shown

---

[*]Equal contribution.

[†]Work done while Yimeng Chen was interning at Institute for AI Industry Research, Tsinghua University.

[‡]Corresponding author.

35th Conference on Neural Information Processing Systems (NeurIPS 2021).

promising improvements on the out-of-distribution performance. EBD methods, e.g., PoE [8], DRiFt [15], and Inverse-Reweight [39], usually adopt a two-stage framework. Firstly, a biased predictor is trained based on the bias features only, namely the *bias-only model*. Its output is then utilized to adjust the learning target of the *main model* by using different ensembling strategies. Previous works are mainly limited to designing different ensembling strategies, without considering the bias-only model, which clearly plays an essential role in the whole process.

In this paper, we focus on investigating the bias-only model in the EBD methods. We theoretically reveal that the quality of the predictive uncertainty estimation given by the bias-only model is crucial for the debiasing performance of EBD methods. Specifically, we prove that the out-of-distribution accuracy of the debiased model is monotonically decreasing with the calibration error of the bias-only model when such error exceeds a threshold[4]. Moreover, by theoretically analyzing the decline of in-distribution performance caused by debiasing, we show the existence of the case when uncertainty calibration can also mitigate such a side-effect. Empirically, we show that bias-only models employed by existing methods on both natural language inference and fact verification tasks fail to produce accurate uncertainty estimations. These findings indicate the critical role of the calibration property of current bias-only models for further improvement of EBD methods.

Motivated by the theoretical analysis and empirical study, we introduce an additional calibration stage into the previous EBD methods. In this stage, the bias-only model is calibrated with model-agnostic calibration methods to obtain more accurate predictive uncertainty estimation. Specifically, two typical calibration methods are used in this paper, i.e. temperature scaling [12] and Dirichlet calibration [19]. After that, the calibrated bias-only model is used to train the main model with off-the-shelf ensembling strategies. In this way, we extend the traditional two-stage EBD framework to a three-stage one, including bias **Mo**deling, model **Ca**librating, and **D**ebiasing, named MoCaD for short.

To demonstrate the effectiveness of our proposed framework, we conduct experiments on four challenging benchmarks for two NLU tasks, i.e. natural language inference and fact verification. Experimental results show that our framework significantly improves the out-of-distribution performance, as compared with the traditional two-stage one. Moreover, our theoretical results are well verified by the empirical observations in real scenarios.

Our main contributions can be summarized as the following three folds.

- We explore, both theoretically and empirically, the effect of the bias-only model in the EBD methods. Consequently, a critical problem is revealed: existing bias-only models are poorly calibrated, which will hurt the debiasing performance.
- We propose a model-agnostic three-stage EBD framework to tackle the above problem.
- We conduct extensive experiments on four challenging datasets for two different tasks, and experimental results show the superiority of our proposed framework as against the traditional two-stage one.

## 2   Related Work

**Dataset Bias.**   Various biases have been found in different NLU benchmarks. For example, models with partial input can perform much better than majority-class baselines in NLI and fact verification datasets [13; 27; 30]. Many multi-hop questions can be solved by just using single-hop models in the recent multi-hop QA datasets [24; 6]. Similar phenomena have been observed in many other tasks, such as reading comprehension [18] and visual question-answering [1]. Many models have used such superficial cues to achieve remarkable performance instead of capturing the underlying intrinsic principles in these biased datasets, leading to poor generalization on out-of-distribution datasets, when the relation of bias features and labels are changed [23; 30; 21].

**Ensemble-based debiasing (EBD) methods.**   EBD methods are a kind of model-agnostic debiasing method to reduce the reliance of models on specific dataset bias. In these methods, a bias-only

---

[4]This condition is more general when the ground truth labeling based on the signal features has low certainty. Such cases exist in natural language understanding (NLU) tasks, where the ground-truth label for a sample is not unique but inherently forms a distribution, as shown by recent empirical studies [26; 25]. That is why we focus our empirical study on NLU tasks.

model is used to assist the debiasing training of the main model. Most EBD methods, e.g., PoE [8], DRiFt [15], and Inverse-Reweight [39], can be formalized as a two-stage framework. It is commonly assumed that the dataset bias is known a-priori. In the first stage, the bias-only model is trained to capture the dataset bias by leveraging the pre-defined bias features. Then the bias-only model is used to adjust the learning target of the main model with different ensembling strategies. Recently, some works start to improve the EBD methods by exploring the bias-only models. For example, Utama et al. [35], Sanh et al. [29], and Clark et al. [9] focus on relaxing the basic assumption of many EBD methods, i.e., the dataset bias is known a-priori. They exploit different prior knowledge to obtain bias-only models, e.g., models that shallow [35] or with limited capacity [29; 9] are considered to be biased. Unlike these works, we theoretically study the essential effect of the bias-only model on the final debiasing performance and show how to improve it in the algorithm design process. Please note that some works [22; 9] have been proposed to jointly learn the bias-only model and the debiased main model in an end-to-end manner. However, Since it is difficult to quantify the impact of the bias-only model in this scheme, we mainly focus on the typical two-stage methods [8; 15; 39; 35; 29].

## 3 Formalization of EBD Methods

In this section, we formalize EBD methods with an introduction to some related notations. Consider a general classification task, where the target is to map an input value $x \in \mathcal{X}$ of an input random variable $X$ to a target label $y \in \mathcal{Y}$ of a target random variable $Y$. We denote features of $x$ that have invariant relations with the label as signal $x^s$, e.g., the sentiment words in sentiment analysis. Conversely, features whose correlation with label $Y$ is spurious and prone to change in the out-of-distribution setting are denoted as bias $x^b$, e.g., the length of input sentences in the NLU tasks. The corresponding random variables are respectively denoted as $X^S$ and $X^B$. Now suppose that on a training dataset $\mathcal{D}$ where $(X, Y) \sim \mathbb{P}_\mathcal{D}(\mathcal{X} \times \mathcal{Y})$, $X^B$ and $Y$ are spuriously correlated. The goal of debiasing is to learn a classifier that models $\mathbb{P}_\mathcal{D}(Y|X^S)$ with invariant out-of-distribution performance.

The following decomposition forms the theoretical basis for EBD methods: for $\forall x \in \mathcal{X}$, with its corresponding features $X^B = x^b$, $X^S = x^s$,

$$\mathbb{P}_\mathcal{D}(Y|X = x) \propto \mathbb{P}_\mathcal{D}(Y|X^B = x^b)\mathbb{P}_\mathcal{D}(Y|X^S = x^s)\frac{1}{\mathbb{P}_\mathcal{D}(Y)}, \tag{1}$$

where $\mathbb{P}_\mathcal{D}(Y|X^B = x^b)$ is the conditional probability distribution of $Y$ given the value of bias features $X^B$, $\mathbb{P}_\mathcal{D}(Y|X^S = x^s)$ represents the true principle we would like to learn, and $\mathbb{P}_\mathcal{D}(Y|X = x)$ is the conditional distribution of $Y$ given all features, which is usually approximated by directly applying statistical machine learning methods on the training data. This decomposition can be deduced under the constraint that $X^S \perp\!\!\!\perp X^B|Y$, as shown in [8; 15; 9]. We further prove that it also holds with the assumptions in [39] (See the appendix). The theoretical analysis in this paper is conducted based on the same constraint as in [8; 15; 9].

From this decomposition, the true principle $\mathbb{P}_\mathcal{D}(Y|X^S)$ can be achieved by adjusting the learning target with $\mathbb{P}_\mathcal{D}(Y|X^B)$. This is exactly the basic idea of EBD methods.

Most EBD methods belong to a two-stage framework. In the first stage, a bias-only model $f_B : \mathcal{X} \to \mathbb{R}^{|\mathcal{Y}|}$ is trained to approximate $\mathbb{P}_\mathcal{D}(Y|X^B)$. Then it is employed to adjust the learning target in a direct or indirect way. Direct methods such as Inverse-Reweight [39] reweight the distribution by the inverse of the probability induced by the bias-only model to approximate the true principle. The objective function of the main model $f_M : \mathcal{X} \to \mathbb{R}^{|\mathcal{Y}|}$ becomes:

$$\min_{f_M} \mathbb{E}_{X,Y \sim \mathbb{P}_\mathcal{D}}\left[\frac{1}{p_Y^b(X)}\mathcal{L}_c(Y, \mathbf{p}^m(X))\right], \tag{2}$$

where $\mathbf{p}^b(X) = \{p_1^b(X), p_2^b(X), \ldots, p_{|\mathcal{Y}|}^b(X)\}$, $\mathbf{p}^m(X) = \{p_1^m(X), p_2^m(X), \ldots, p_{|\mathcal{Y}|}^m(X)\}$ denote the uncertainty estimations, i.e. the prediction probabilities given by $f_B$ and $f_M$ respectively. $\mathcal{L}_c$ represents the cross-entropy loss function. On the other hand, indirect methods usually utilize the output of the bias-only model to adjust the loss function of the main model, and the learning target becomes:

$$\min_{f_M} \mathbb{E}_{X,Y \sim \mathbb{P}_\mathcal{D}}[\mathcal{L}_c(Y, m(\mathbf{q}^b(X) \cdot \mathbf{q}^m(X)))], \tag{3}$$

where $m$ is the normalization function, and $\mathbf{q}^b(X), \mathbf{q}^m(X)$ are vectors in proportion to $\mathbf{p}^b(X)$ and $\mathbf{p}^m(X)$ respectively. Specifically, PoE [8; 35] directly uses the probability output, DRiFt [15] and Sanh et al. [29] utilizes exponential of the logits output. In Learned-Mixin [8], a variant of PoE, $\mathbf{q}^b(X)$ is changed to $(\mathbf{p}^b(X))^{g(X)}$, where $g(X)$ is a trainable gate function.

For both direct and indirect methods, by the property of the cross-entropy loss [14], the optimal main model $f_M^*$ satisfies $\mathbf{p}^{m*} \propto \mathbb{P}_{\mathcal{D}}(Y|X)/\mathbf{p}^b$. Therefore, we have $\mathbf{p}^{m*} \propto \mathbb{P}_{\mathcal{D}}(Y|X^S)$ when $\mathbf{p}^b \propto \mathbb{P}_{\mathcal{D}}(Y|X^B)$, which guarantees the effectiveness of the existing EBD methods. Please note that Learned-Mixin does not satisfy this property due to the trainable gate function.

## 4 Analysis of the Bias-only Model

Bias-only models are critical to EBD methods, since their outputs are used to help recover the unbiased distribution. However, far too little attention has been paid to them in previous research. In this section, we theoretically quantify the effect of bias-only outputs on the final debiasing performance and empirically show the weakness of existing bias-only models.

### 4.1 Theoretical Analysis

According to the discussion in Section 3, the optimal main model $f_M^*$ induces the following conditional probability:

$$\mathbb{P}_{\mathcal{D}, f_M^*}(Y=i|X) := \frac{\mathbb{P}_{\mathcal{D}}(Y=i|X)/p_i^b(X)}{\sum_{j \in \mathcal{Y}} \mathbb{P}_{\mathcal{D}}(Y=j|X)/p_j^b(X)}. \tag{4}$$

For arbitrary $x \in \mathcal{X}$, we define

$$Y(x) := \operatorname{argmax}_{i \in \mathcal{Y}} \mathbb{P}_{\mathcal{D}}(Y=i|X^S=x^s), \tilde{Y}(x) := \operatorname{argmax}_{i \in \mathcal{Y}} \mathbb{P}_{\mathcal{D}, f_M^*}(Y=i|X=x). \tag{5}$$

Here $Y(x)$ stands for the predicted label given by the intrinsic principle, and $\tilde{Y}(x)$ is the label prediction given by the debiased main model. With these notations, the debiasing performance can be defined as $\mathbb{E}_{X \sim \mathbb{P}_{\mathcal{D}}(X)}(\tilde{Y}(X) = Y(X))$. As the major factor related to the bias-only model is $p_i^b(X)$, i.e. the uncertainty estimation, in the concerned quantities $\tilde{Y}(X)$, we investigate the effect of the bias-only model on the debiasing performance from this aspect.

Without loss of generality, we consider the binary classification problem with $\mathcal{Y} = \{0, 1\}$ and balanced label distribution. To divide and conquer, we conduct the theoretical analysis on a set of samples, where the bias-only model generates the same uncertainty estimation, i.e. $\mathcal{S}_{f_B}(l) := \{x|p_0^b(x) = l\}, \forall l \in [0, 1]$. Specifically, the quality of the uncertainty estimation of the bias-only model on $\mathcal{S}_{f_B}(l)$ can be measured by the calibration error defined as $|l - \mathbb{P}_{\mathcal{D}}(Y=0|\mathcal{S}_{f_B}(l))|$. The debiasing performance on $\mathcal{S}_{f_B}(l)$ is defined as $\mathbb{P}_{\mathcal{D}}(\{x \in \mathcal{S}_{f_B}(l)|\tilde{Y}(x) = Y(x)\})$, i.e. the probability of the subset of $\mathcal{S}_{f_B}(l)$ on which the main model gives the same prediction as the intrinsic principle.

The following theorem formalizes a precise result. Specifically, the debiasing performance is a monotonically decreasing function of the calibration error when it exceeds a deviation threshold $\delta(l_0, \epsilon, \alpha)$. Here $\alpha := \min_{X^S} \max_{i \in \{0,1\}} \mathbb{P}_{\mathcal{D}}(Y=i|X^S)$ denotes the global certainty level of the true principle $\mathbb{P}_{\mathcal{D}}(Y|X^S)$.

**Theorem 1.** *For any $l \in [0, 1]$, assume that $\exists l_0$ s.t. $\mathbb{P}_{\mathcal{D}}(Y=0|X^B) \in (l_0 - \epsilon, l_0 + \epsilon)$ when $X$ takes values in $\mathcal{S}_{f_B}(l)$. If the calibration error $|l - \mathbb{P}_{\mathcal{D}}(Y=0|\mathcal{S}_{f_B}(l))| \geq \delta(l_0, \epsilon, \alpha) > 0$, the debiasing performance $\mathbb{P}_{\mathcal{D}}(\{x \in \mathcal{S}_{f_B}(l)|\tilde{Y}(x) = Y(x)\})$ declines as $|l - \mathbb{P}_{\mathcal{D}}(Y=0|\mathcal{S}_{f_B}(l))|$ increases, where $\delta(l_0, \epsilon, \alpha)$ is a constant dependent with $l_0$, $\epsilon$ and $\alpha$. When $\alpha < \frac{1}{2} + \frac{\epsilon}{2l_0(1-l_0)+2\epsilon^2}$, $0 \leq \delta(l_0, \epsilon, \alpha) < 2\epsilon$, where $2\epsilon \leq \frac{\epsilon}{2l_0(1-l_0)+2\epsilon^2} < \frac{1}{2}$. Otherwise $C < \delta(l_0, \epsilon, \alpha) < 2\epsilon + C$, where $0 < C := l_0 - \epsilon - \frac{l_0+\epsilon}{(l_0+\epsilon)+(1-l_0-\epsilon)\frac{\alpha}{1-\alpha}}$, which increases as $\alpha$ increases.*

The threshold in this theorem depends on latent constants $l_0$, $\epsilon$, and $\alpha$. Here $l_0$ and $\epsilon$ define the range of $\mathbb{P}_{\mathcal{D}}(Y=0|X^B)$ on $\mathcal{S}_{f_B}(l)$. As these constants are related to the posterior characteristics of $f_B$, we verify the generality of such condition by empirical facts in Section 6. Note that the deviation threshold decreases as the certainty level $\alpha$ decreases. That means the same calibration error is more likely to exceed the threshold under smaller $\alpha$, resulting in a more considerable decrease in debiasing

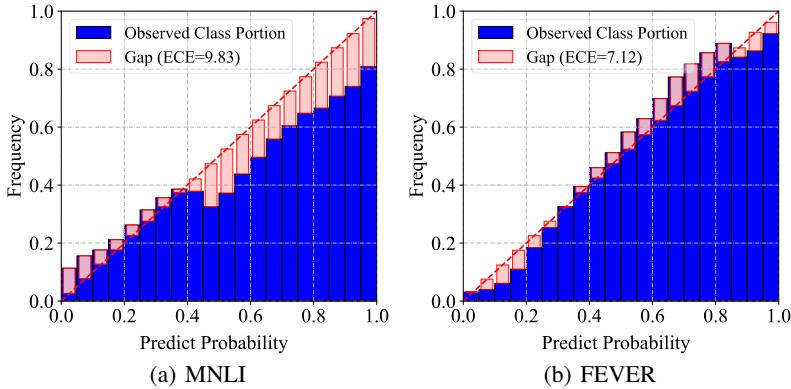

Figure 1: Reliability diagrams of the bias-only models on MNLI and FEVER. The x-axis is the predictive probability of the bias-only model, and the y-axis is the frequency. The wide blue bars show the weighted average of the observed class portion to all classes within each bin, and the narrow red bars show the gap between the observed class portion and the predictive probability of the bias-only model.

performance. As a result, the condition in Theorem 1 is more general and significant when the true principle $\mathbb{P}_{\mathcal{D}}(Y|X^S)$ has low certainty, for example, in the NLU tasks as supported by empirical evidence in [26; 25].

We also theoretically analyze the effect of the bias-only model on the in-distribution performance, which is defined as $\mathbb{E}_{X \sim \mathbb{P}_{\mathcal{D}}(X)}(\tilde{Y}(X) = \hat{Y}(X))$, where $\hat{Y}(x) := \operatorname{argmax}_{i \in \mathcal{Y}} \mathbb{P}_{\mathcal{D}}(Y = i|X = x)$ denotes the label given by the ideal predictor on $\mathcal{D}$. The result is shown in the following theorem.

**Theorem 2.** *For any $X$, $\tilde{Y}(X) \neq \hat{Y}(X)$ if and only if $p^b_{\hat{Y}(x)}(x) > \mathbb{P}_{\mathcal{D}}(Y = \hat{Y}(x)|X = x)$.*

Theorem 2 gives a possible explanation for the decrease of in-distribution performance of EBD debiased models: the in-distribution error occurs when the predictive uncertainty estimation of the bias-only model on $\hat{Y}(x)$ is higher than the conditional probability of $\hat{Y}(x)$. That indicates that the in-distribution error is non-decreasing as the range of the uncertainty estimation of bias-only models increases. As an important case, when the bias-only model is over-confident [12], decreasing its calibration error can improve both the in-distribution and out-of-distribution performance of the debiased model according to the two theorems.

To sum up, our theoretical study shows that both debiasing and in-distribution performances of the EBD methods are affected by the uncertainty estimation of the bias-only models. Please note that both Theorem 1 and 2 can be generalized to multi-class scenarios, with a more complex form. For simplicity, we only discuss the binary class case.

### 4.2 Empirical Analysis

According to some recent machine learning studies, the uncertainty estimations of many widely used machine learning classifiers are not reliable [20; 12; 32; 36]. This indicates that the existing bias-only classifiers may fail to produce a good uncertainty estimation, which can hurt the debiasing performance, as demonstrated by our theoretical results.

To quantify the effect, we further conduct an empirical study to demonstrate the quality of the existing bias-only models with respect to the uncertainty estimation. Specifically, we experiment on two typical public datasets, MNLI and FEVER. Their experimental settings and detailed analysis can be found in Section 6.1. For MNLI, we consider the syntactic bias [23] and use hand-crafted features to train a bias-only model, the same as in [8]. For FEVER, we consider the claim-only bias [30] and train a claim-only model as the bias-only model, as in [34]. After that, we use the classwise reliability diagram [19] to check its calibration error based on data binning. We adopt the classwise expected calibration error [19] as a measure to quantify the quality of the uncertainty estimation, denoted as ECE for short, with its lower value indicates better-calibrated uncertainty estimation.

Now we introduce our experimental results. The classwise reliability diagrams on MNLI and FEVER training sets are plotted in Figure 1(a) and Figure 1(b), respectively. For perfectly calibrated predictions, the curve in a reliability diagram should be as close as possible to the diagonal. Therefore, the deviation from the diagonal represents the calibration error. From the results, we can see that existing bias-only models suffer from inaccurate uncertainty estimation problems on both datasets.

# 5 The MoCaD Framework

To overcome the unreliable predictive uncertainty problem, we introduce a calibration operation to the bias-only model, achieving a **Mo**deling, **Ca**librating and **D**ebiasing framework, named MoCaD for short. Our framework consists of three stages. Firstly, we train a bias-only model to model $\mathbb{P}_{\mathcal{D}}(Y|X^B)$. Secondly, we use the model-agnostic calibration methods to improve the calibration error of the bias-only model. The calibrated bias-only model is finally employed to conduct the debiasing process through the existing ensembling strategies.

## 5.1 Bias Modeling

In the first stage, we train a bias-only model to approximate $\mathbb{P}_{\mathcal{D}}(Y|X^B)$, similar to previous works [8; 15]. When the dataset bias is identified, i.e. bias features $X_B$ are known a-priori [8; 15], the bias-only model can be obtained by only using the pre-defined $X_B$ to predict label $y$ with cross-entropy loss. For example, in NLI, many specific linguistic phenomena in hypothesis sentences such as negation are highly correlated with certain inference classes [27]. In this case, hypothesis sentences are used as inputs to train an NLI model as a bias-only model. When the dataset bias is unknown, a 'shallow' model or a 'weak' model can be built as the bias-only model, as in [35; 29].

## 5.2 Model Calibrating

We propose to utilize model-agnostic calibration methods to improve the calibration error of the bias-only models. Specifically, two typical calibration methods, temperature scaling [12] and Dirichlet calibrator [19], are used in this paper. The calibrated bias-only model is denoted as $\tilde{f}_B$.

Temperature scaling is a simple-but-effective calibration method. It learns a single scalar parameter 'temperature' which is applied to the last softmax layer. Specifically, denote $\mathbf{z}^b(X)$ as the logit output of the bias-only model on sample $(X, Y)$, abbreviated for $\mathbf{z}^b$, temperature scaling will correct the output as follows: $\tilde{\mathbf{p}}^b = \mathrm{softmax}(\mathbf{z}^b/T)$, where $T$ is the temperature, which is learned with the cross-entropy loss.

Dirichlet calibrator is derived from the Dirichlet distribution likelihood. The transformed probability is computed as $\tilde{\mathbf{p}}^b = \mathrm{softmax}(\mathbf{W} \ln \mathbf{p}^b + \mathbf{b}')$, where $\mathbf{W}$ and $\mathbf{b}'$ stand for the linear transformation matrix and intercept term, which are optimized by the cross-entropy loss equipped with ODIR (Off-Diagonal and Intercept Regularisation) to prevent over-fitting [19].

Please note that temperature scaling does not change the predicted label because the maximum of the softmax function remains unchanged. In other words, it only changes the uncertainty estimation and maintains the model's accuracy. Unlike temperature scaling, the Dirichlet calibrator can change the prediction accuracy. Empirically, we observed that the Dirichlet calibrator improves the accuracy of all bias-only models in our experiments (See the Appendix for details). In both methods, the calibration error is expected to be reduced by learning the parameters with the cross-entropy loss.

## 5.3 Debiasing

The final step is to train the main model $f_D$ with the calibrated bias-only model $\tilde{f}_B$. Specifically, $\tilde{f}_B$ is applied with the existing ensembling strategies to make the main model $f_D$ approximate the true principle $\mathbb{P}_{\mathcal{D}}(Y|X^S)$, by adjusting the learning target of the main model, as described in Section 3. The design of the main model is highly dependent on the concerned task, as indicated by previous works. For example, a BERT-based classifier is usually used in NLI [15], and a BottomUp-TopDown VQA model is usually adopted in VQA [8].

# 6 Experiments

In this section, we conduct experiments on different real-world datasets to answer two questions: (1) whether our proposed MoCaD framework improves the debiasing performance of the EBD methods; (2) whether the experimental results are consistent with the theoretical findings.

## 6.1 Experimental Settings

We describe our experimental settings, including datasets, models and some training details. More details are provided in the Appendix.

**Datasets and bias-only models.**  We conduct experiments on both fact verification and natural language inference, which are commonly used tasks in debiasing [8; 34; 35]. We follow these works to choose the datasets and design the bias-only models.

Fact verification requires models to validate a claim in the context of evidence. For this task, we use the training dataset provided by the FEVER challenge [31]. The processing and split of the dataset into training/development set are conducted following Schuster et al. [30][5]. It has been shown that FEVER has the claim-only bias, where claim sentences often contain words highly indicative of the target label [30]. So the bias-only model is trained to predict labels by only using claim sentences. Finally, Fever-Symmetric datasets [30] (both version 1 and 2) are used as the test sets for evaluation.

Natural language inference aims to infer the relationship between premise and hypothesis. Recent studies have shown that various biases exist in the widely used NLI datasets [27; 13; 23]. In this paper, we conduct our experiments on MNLI [37] and consider both known bias and unknown bias. For known bias, firstly, we consider the syntactic bias, e.g. the lexical overlap between premise and hypothesis sentences is strongly correlated with the entailment label [23]. So the bias-only model is a classifier using hand-crafted features indicating how words are shared between the two sentences as the input, the same as that in [8]. Finally, HANS (Heuristic Analysis for NLI Systems) [23] is utilized as the challenging dataset for evaluation. Then we consider the hypothesis-only bias, which means that we can only use the hypothesis to predict the relation between premise and hypothesis. So the bias-only model is defined as a classifier trained to predict labels by only using hypothesis. In the experiment, we still use MNLI as the training set and employ two hard MNLI datasets [13; 21] for evaluation. The 'hard' subsets are derived from the MNLI Mismatched dataset with two different strategies: (1) a neural classifier is trained on hypothesis sentences and the wrongly classified instances are treated as 'hard' instances. (2) patterns in hypothesis sentences that are highly correlated to the specific labels are extracted as surface patterns, and samples which against those surface patterns' indications are recognized as 'hard' samples. Therefore, the two challenging dataset are referred to as Hard-CD (Classifier Detected) and Hard-SP (Surface Pattern), corresponding to their creation strategies. For unknown bias, following Utama et al. [35], we build a 'shallow' model as the bias-only model, which has the same architecture as the main model and is trained on a subset of the MNLI training set. Then we use HANS as the challenging dataset for evaluation as Utama et al. [35].

**Baselines and configurations.**  We experiment with 8 implementations of MoCaD, i.e. two different calibrators combined with four different ensembling strategies. The two calibrators are temperature scaling and Dirichlet calibrator, and the four ensembling strategies are those in Product-of-Experts (PoE), Learned-Mixin (LMin), DRiFt, and Inverse-Reweight (Inv-R). We compare the performances of these implementations with their corresponding two-stage EBD methods. We denote different implementations of MoCaD by the name of corresponding EBD methods with the calibrator name as the subscript. We use `TempS` and `Dirichlet` to denote the implemented methods with temperature scaling and Dirichlet as the calibrator, respectively.

In our experiments, we adopt the BERT-based classifier as the main model and follow the standard setup for sentence pair classification [10]. The cross-entropy trained model (denoted as CE) is also included as a baseline, to show the difference between the debiased and un-debiased model. To tackle the high performance variance on challenging datasets as observed by Clark et al. [8], we run each experiment five times and report the mean scores and the standard deviations. For each task, we utilize the training configurations that have been proven to work well in previous studies and keep the same bias-only model for all methods. For Learned-Mixin, the entropy term weight is set to the value

---

[5]https://github.com/TalSchuster/FeverSymmetric

Table 2: Classification accuracy on MNLI.

| Method | Syntactic Bias | | Hypothesis-only Bias | | | Unknown Bias | |
| --- | --- | --- | --- | --- | --- | --- | --- |
| | ID | HANS | ID | $Hard_{CD}$ | $Hard_{SP}$ | ID | HANS |
| CE | 84.2 ± 0.2 | 61.2 ± 3.2 | 84.2 ± 0.2 | 76.8 ± 0.4 | 72.6 ± 2.0 | 84.2 ± 0.2 | 61.2 ± 3.2 |
| PoE | 82.8 ± 0.4 | 68.1 ± 3.4 | 83.2 ± 0.2 | 79.4 ± 0.4 | 76.8 ± 2.4 | 80.7 ± 0.2 | 69.0 ± 2.4 |
| **PoE**$_{TempS}$ | 83.9 ± 0.3 | 69.1 ± 2.8 | 82.9 ± 0.3 | 79.6 ± 0.4 | 77.4 ± 2.4 | 82.1 ± 0.2 | 69.9 ± 1.6 |
| **PoE**$_{Dirichlet}$ | 84.1 ± 0.3 | **70.7** ± 1.5 | 82.7 ± 0.4 | 79.4 ± 0.2 | **77.6** ± 2.1 | 82.3 ± 0.3 | **70.7** ± 1.0 |
| DRiFt | 81.8 ± 0.4 | 66.5 ± 4.0 | 83.5 ± 0.4 | 79.5 ± 0.6 | 76.3 ± 1.6 | 80.2 ± 0.3 | 69.1 ± 1.3 |
| **DRiFt**$_{TempS}$ | 83.0 ± 0.4 | 69.7 ± 1.8 | 83.1 ± 0.2 | 79.6 ± 0.2 | 77.4 ± 3.3 | 81.5 ± 0.3 | **70.0** ± 0.9 |
| **DRiFt**$_{Dirichlet}$ | 83.6 ± 0.3 | **69.8** ± 1.9 | 82.8 ± 0.3 | 79.6 ± 0.2 | **79.0** ± 1.6 | 81.9 ± 0.6 | 69.4 ± 1.1 |
| InvR | 82.5 ± 0.1 | 68.4 ± 1.2 | 83.1 ± 0.2 | 78.4 ± 0.5 | 77.1 ± 2.0 | 78.7 ± 4.8 | 64.7 ± 2.6 |
| **InvR**$_{TempS}$ | 83.6 ± 0.2 | 69.4 ± 1.6 | 82.8 ± 0.2 | 78.6 ± 0.2 | 77.9 ± 1.7 | 81.4 ± 0.5 | 65.8 ± 0.9 |
| **InvR**$_{Dirichlet}$ | 83.7 ± 0.4 | **69.4** ± 1.3 | 82.5 ± 0.2 | 78.9 ± 0.4 | **80.8** ± 2.0 | 81.5 ± 0.2 | **68.2** ± 0.8 |
| LMin | 84.1 ± 0.3 | **65.5** ± 3.7 | 80.5 ± 0.3 | 80.0 ± 0.4 | 78.2 ± 2.0 | 83.1 ± 0.3 | **66.5** ± 1.1 |
| **LMin**$_{TempS}$ | 84.1 ± 0.2 | 63.2 ± 2.7 | 80.5 ± 0.6 | 80.3 ± 0.2 | 80.8 ± 3.6 | 83.3 ± 0.2 | 66.2 ± 1.0 |
| **LMin**$_{Dirichlet}$ | 84.3 ± 0.3 | 62.7 ± 2.6 | 80.1 ± 0.5 | 79.8 ± 0.4 | **83.2** ± 2.2 | 82.7 ± 0.2 | 66.4 ± 1.2 |

suggested by Utama et al. [34]. For the Dirichlet calibrator, we set $\lambda = 0.06$ for all experiments, based on the in-distribution performance on the development sets.

## 6.2 Experimental Results

Now we show our experimental results to answer the aforementioned two questions.

Table 1 shows the experimental results on FEVER. We can see that for both calibrators, MoCaD outperforms the corresponding EBD methods, including Learned-Mixin, on both Fever-Symmetric v1 and v2 datasets. Comparing different calibrators, `Dirichlet` consistently performs better than `TempS`. Please note that the label distribution of the development set is different from that of the training set on FEVER, which explains why sometimes `Dirichlet` obtains better in-distribution performance than the cross-entropy loss.

Table 2 shows the experimental results on MNLI with respect to known bias and unknown bias. The main results are similar to that on FEVER, i.e. calibration brings benefit to the debiasing performance, and `Dirichlet` obtains better results than `TempS`, for all EBD methods except Learnd-Mixin on HANS. It indicates that for both known and unknown dataset bias, MoCaD outperforms corresponding EBD methods. Please note that, as a trainable gate function is added in Learned-Mixin, the optimal bias-only model of it is different from others and does not fit our theoretical assumptions. Specially, the performance gap between baselines and our methods is relatively small on Hard-CD. This may due to the fact that the construction of Hard-CD is dependent on a specific biased model.

Table 1: Classification accuracy on FEVER.

| Method | ID | Symm. v1 | Symm. v2 |
| --- | --- | --- | --- |
| CE | 87.1 ± 0.6 | 56.5 ± 0.9 | 63.9 ± 0.9 |
| PoE | 84.0 ± 1.0 | 62.0 ± 1.3 | 65.9 ± 0.6 |
| **PoE**$_{TempS}$ | 82.0 ± 0.9 | 63.3 ± 0.9 | 66.4 ± 0.8 |
| **PoE**$_{Dirichlet}$ | 87.1 ± 1.0 | **65.9** ± 1.1 | **69.1** ± 0.8 |
| DRiFt | 84.2 ± 1.2 | 62.3 ± 1.5 | 65.9 ± 0.7 |
| **DRiFt**$_{TempS}$ | 81.7 ± 0.9 | 63.5 ± 1.3 | 66.5 ± 0.7 |
| **DRiFt**$_{Dirichlet}$ | 87.4 ± 1.2 | **65.7** ± 1.4 | **69.0** ± 1.3 |
| InvR | 84.3 ± 0.8 | 60.8 ± 1.2 | 65.2 ± 1.0 |
| **InvR**$_{TempS}$ | 83.8 ± 0.6 | 61.5 ± 0.9 | 65.4 ± 0.7 |
| **InvR**$_{Dirichlet}$ | 87.0 ± 0.8 | **63.8** ± 2.2 | **68.2** ± 1.7 |
| LMin | 84.7 ± 1.8 | 59.8 ± 2.7 | 65.3 ± 1.1 |
| **LMin**$_{TempS}$ | 84.9 ± 1.7 | 60.0 ± 2.5 | 65.6 ± 1.5 |
| **LMin**$_{Dirichlet}$ | 87.5 ± 1.1 | **61.5** ± 2.4 | **67.1** ± 1.3 |

### 6.2.1 Empirical Verification of Theorem 1

Now we analyze whether the improvement of debiasing performance agrees with our theoretical study in Theorem 1. That is, calibrated models achieve better uncertainty estimation, leading to better debiasing performance results.

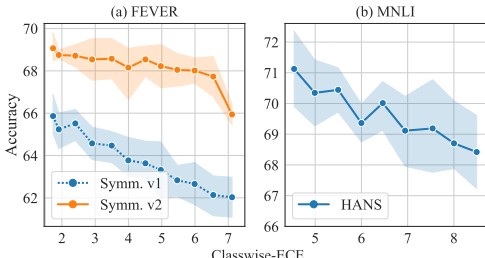
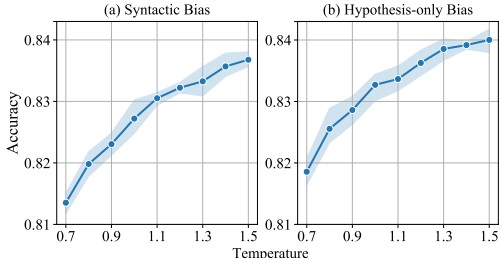

Figure 2: Debiasing performance of bias-only model vs the quality of predictive uncertainty measured by classwise-ECE (lower is better).

Figure 3: In-distribution performance (accuracy) of the main model vs temperature.

To facilitate the study, we demonstrate the classwise-ECE of the calibrated bias-only models on different training datasets, as shown in Table 3. In the table, Un-Cal, Dirichlet, and TempS denote the bias-only model without calibration, with temperature scaling and Dirichlet calibrator, respectively. From the results, we can see that calibrated bias-only models on different datasets achieve better uncertainty estimation, for both calibrators. Comparing the two calibrators, the Dirichlet calibrator performs better because of its higher expressive power. Further considering the debiasing improvement in Table 1 and 2, we can see that the empirical findings consist with our theory.

Furthermore, we conduct a more detailed experiment on MNLI and FEVER, regarding syntactic bias and claim-only bias respectively. Specifically, we adopt the ensembling strategy in PoE, and calibrate bias-only models with the Dirichlet calibrator and save models at different checkpoints. Then we consider the debiasing performances of bias-only models with different uncertainty estimation qualities, measured by classwise-ECE. The results are plotted in Figure 2. We can see that when the classwise-ECE grows, i.e. the calibration error of the bias-only model grows, the accuracy on the test set decreases, i.e. the debiasing performance drops. These results precisely prove Theorem 1.

### 6.2.2 Empirical Verification of Theorem 2

Theorem 2 reveals the relation between the confidence of the bias-only model and the in-distribution error of the main model. That is, if the confidence, i.e. the uncertainty estimation of the bias-only model on the predicted label is reduced, the in-distribution error of the main model will decrease. Since the label distribution changes on the development set of FEVER, we only consider the results on MNLI. From Ta-

Table 3: Classwise-ECE of the calibrated bias-only models on different training datasets.

|           | FEVER | HANS | MNLI | Unknown |
|-----------|-------|------|------|---------|
| Un-Cal    | 7.11  | 9.83 | 3.01 | 7.41    |
| TempS     | 6.23  | 7.70 | 2.38 | 3.07    |
| Dirichlet | 1.73  | 4.47 | 0.87 | 1.45    |

ble 2, the in-distribution performance increases in the scenario of syntactic and unknown bias and decreases in the scenario of hypothesis-only bias, for most implementations. That is because the syntactic and unknown bias-only model is over-confident, and the hypothesis-only bias-only model is under-confident, as shown in our Appendix. These results are accordant with our theory.

We provide a detailed experiment to further explain the relationship revealed in Theorem 2. Specially, we adopt the ensembling strategy in PoE and take temperature scaling as the calibrator, because the temperature parameter controls the confidence of the calibrated model. The bigger the temperature, the less confident the obtained model. We manually set the temperature parameter from 0.7 to 1.5 with step 0.1, and record the in-distribution accuracy on the development set for the calibrated bias-only model with PoE. The results are plotted in Figure 3. It shows that when the bias-only model is less confident, the in-distribution performance of the main model improves, which verifies Theorem 2.

## 7 Conclusions and Future Work

This paper theoretically and empirically reveals an important problem, which is ignored in previous studies, that existing bias-only models in the EBD methods are poor-calibrated, leading to unsatis-

factory debiasing performances. To tackle this problem, we propose a three-stage EBD framework (MoCaD), including bias modeling, model calibrating, and debiasing. Extensive experiments on natural language inference and fact verification tasks show that MoCaD outperforms corresponding EBD methods, regarding known and unknown dataset bias. Furthermore, our detailed empirical analyses verify the correctness of our theorems. We believe that our study will draw people's attention to the bias-only model, which has the potential to become an interesting research direction in the debiasing study. A limitation of this paper is that our empirical studies focus on NLU tasks. Further experimental results on image classification show inconsistent improvements (See the appendix). A possible reason is that image classes (e.g., birds or elephants) are less disputed than language concepts (e.g., entailment or neutral). Thus the invariant mechanism for image classification has a higher certainty, reducing the impact of calibration error on debiasing according to our theoretical analysis. In the future, we plan to extend our investigations to end-to-end EBD methods and more tasks besides NLU.

## Acknowledgments and Disclosure of Funding

This work is supported by the National Key R&D Program of China under Grants No. 2020AAA0105200, the National Natural Science Foundation of China (NSFC) under Grants No. 61773362, and 61906180.

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
