# Appendix: Uncertainty Calibration for Ensemble Based Debiasing Methods

**Ruibin Xiong**[1,2,3*], **Yimeng Chen**[2,4*†], **Liang Pang**[2,5], **Xueqi Cheng**[1,2],
**Zhiming Ma**[2,4] and **Yanyan Lan**[6‡]

[1]CAS Key Laboratory of Network Data Science and Technology,
Institute of Computing Technology, Chinese Academy of Sciences
[2]University of Chinese Academy of Sciences [3]Baidu Inc.
[4]Academy of Mathematics and Systems Science, Chinese Academy of Sciences
[5]Data Intelligence System Research Center,
Institute of Computing Technology, Chinese Academy of Sciences
[6] Institute for AI Industry Research, Tsinghua University
{xiongruibin18, chenyimeng14}@mails.ucas.ac.cn,
{cxq, pangliang}@ict.ac.cn,
mazm@amt.ac.cn,lanyanyan@tsinghua.edu.cn

## A Experiment Settings

### A.1 Experimental Settings on FEVER

**Bias-only Model** The bias-only model is a nonlinear classifier trained on top of the vector representation of the claim sentence. We obtain this vector representation by max-pooling word embeddings into a single vector as in Utama et al. [13].

**Training Details** We follow Schuster et al. [11] to fine-tune the `bert-base-uncased` model using the following configuration:learning rate is set to $2 \times 10^{-5}$ and training for 3 epochs. Early stopping on validation accuracy is adopted. We use 5-fold internal cross-validation to train the Dirichlet calibrator and ensemble these calibrators by averaging their predictions. For the Dirichlet calibrator, we drop the bias term, and consider $\lambda \in \{0.03, 0.06, 0.003, 0.006\}$ and set $\lambda = 0.06$ in all experiments, according to the in-distribution performance on the development sets. We use 5-fold internal cross-validation to train the Dirichlet calibrator and ensemble these calibrators by averaging their predictions.

### A.2 Experimental Settings on MNLI

**Bias-only Model** For syntactic bias, we train a nonlinear classifier on top of the hand-crafted features. Following Clark et al. [4], the hand-crafted features include (1) whether all words in the hypothesis exist in the premise; (2) whether the hypothesis is a continuous subsequence of the premise; (3) the fraction of premise words that shared with hypotheses; (4) the mean, min, max of cosine similarities between word vectors in the premise and the hypothesis. We consider the same weight for neutral and contradiction class during training by mapping these labels into non-entailment and divide the outputs of non-entailment during debiasing training. For hypothesis-only bias, we train a nonlinear classifier on top of an LSTM-based sentence encoder, which only uses hypothesis sentence as input to predict the labels, as in Utama et al. [13]. For unknown bias, we build a 'shallow' model as the bias-only model. It is a `bert-base-uncased` model fine-tuned on a subset of MNLI

---

*Equal contribution.

†Work done while Yimeng Chen was interning at Institute for AI Industry Research, Tsinghua University.

‡Corresponding author.

35th Conference on Neural Information Processing Systems (NeurIPS 2021).

training set for 3 epochs using the learning rate of $5 \times 10^{-5}$. The subset contained 2K examples randomly sampled from MNLI training set, as in Utama et al. [14].

**Training Details** For both hypothesis-only bias and syntactic bias, we fine-tune the `bert-base-uncased` model for all settings using the default configuration: learning rate is set to $5 \times 10^{-5}$ and training for 3 epochs, as in Utama et al. [13]. The exception is for DRiFt on syntactic bias since we found it convergences slow on the in-distribution development set. We train it for 6 epochs for all settings. Early stopping on validation accuracy is adopted. For unknown bias, we following Utama et al. [14] to fine-tune `bert-base-uncased` model for all settings using the following configuration: learning rate is set to $5 \times 10^{-5}$ and training for 5 epochs. We observed that MoCaD framework converges faster on the challenging dataset than the original EBD methods. Since the assumption is not having access to any out-of-domain test data, and there is no available development set for HANS, we follow [2; 10] to perform the model section on the test set. Here, we simply pick the model trained at the second-to-last epoch for MoCaD on unknown bias. For the Dirichlet calibrator, we use the same configuration as in FEVER.

## B Proof for Decomposition

*Proof.* In Zhang et al. [15], it is assumed that there exists a *leakage-neutral* distribution $\mathscr{D}$ with domain $\mathcal{X} \times \mathcal{Y} \times \mathcal{L} \times \mathcal{S}$, where $\mathcal{X}$ is the input feature space, $\mathcal{L}$ is the sampling strategy feature space and $\mathcal{S}$ is the binary sampling intention space. The observed distribution is denoted as $\hat{\mathscr{D}}$, which satisfies $\mathbb{P}_{\hat{\mathscr{D}}}(x,y,l) = \mathbb{P}_{\mathscr{D}}(x,y,l|S=Y)$. In the following, we omit the subscripts for $\mathscr{D}$. The following assumptions are adopted in [15]:

$$\mathbb{P}(Y|L) = \mathbb{P}(Y), \tag{1}$$
$$\mathbb{P}(S|X,Y,L) = \mathbb{P}(S|L) \tag{2}$$

In this framework, $\mathbb{P}(Y|X)$ is supposed to be the true principle to learn, corresponding to $\mathbb{P}_{\mathcal{D}}(Y|X^S)$ in our notation. Now we prove the following decomposition

$$\mathbb{P}_{\hat{\mathscr{D}}}(Y|X) \propto \mathbb{P}_{\hat{\mathscr{D}}}(Y|L)\mathbb{P}(Y|X)\frac{1}{\mathbb{P}_{\hat{\mathscr{D}}}(Y)} \tag{3}$$

Correspondingly, by our notations we have

$$\mathbb{P}_{\hat{\mathscr{D}}} = \mathbb{P}_{\mathcal{D}}, L = X^B.$$

As a result, equation 3 is equivalent to the decomposition (1) in the main paper. To prove this equation, firstly,

$$\mathbb{P}_{\hat{\mathscr{D}}}(Y=y|X) = \mathbb{P}(Y=y|X, S=Y)$$
$$= \frac{\mathbb{P}(Y=y, S=y, X)}{\mathbb{P}(X, S=Y)}$$
$$= \frac{\mathbb{P}(S=y|Y=y, L, X^S)\mathbb{P}(Y=y, X)}{\mathbb{P}(X, S=Y)}$$
$$= \mathbb{P}(S=y|L)\mathbb{P}(Y=y|X)\frac{\mathbb{P}(X)}{\mathbb{P}(X, S=Y)}$$
$$\propto \mathbb{P}(S=y|L)\mathbb{P}(Y=y|X)$$

Secondly,

$$\mathbb{P}_{\hat{\mathscr{D}}}(Y=y|L) = \mathbb{P}(Y=y|L, S=Y)$$
$$= \frac{\mathbb{P}(Y=y, S=y, L)}{\mathbb{P}(L, S=Y)}$$
$$= \frac{\mathbb{P}(S=y|Y=y, L)\mathbb{P}(Y=y, L)}{\mathbb{P}(L, S=Y)}$$
$$= \mathbb{P}(S=y|L)\mathbb{P}(Y=y)\frac{\mathbb{P}(L)}{\mathbb{P}(L, S=Y)}$$

By the above equations we have

$$\mathbb{P}_{\hat{\mathscr{D}}}(Y = y | X) \propto \frac{\mathbb{P}_{\hat{\mathscr{D}}}(Y = y | L)}{\mathbb{P}(Y = y)} \mathbb{P}(Y | X) \frac{\mathbb{P}(L, S = Y)}{\mathbb{P}(L)}$$

$$\propto \mathbb{P}_{\hat{\mathscr{D}}}(Y = y | L) \mathbb{P}(Y | X) \frac{1}{\mathbb{P}(Y = y)}$$

As $\mathbb{P}(Y)$ is a prior parameter chosen to balance the posterior distribution, it can be proved that this condition is satisfied when it equals $\mathbb{P}_{\mathcal{D}}(Y)$, as follows:

$$\mathbb{P}_w(Y) \propto \sum_l \frac{\mathbb{P}(Y)}{\mathbb{P}_{\hat{\mathscr{D}}}(Y | L = l)} \mathbb{P}_{\hat{\mathscr{D}}}(Y | L = l) \mathbb{P}_{\hat{\mathscr{D}}}(L = l) = \mathbb{P}(Y)$$

where $\mathbb{P}_w(Y)$ denotes the distribution of $Y$ after the reweighting. As a result $\mathbb{P}_w(Y = y) \propto \mathbb{P}_{\mathcal{D}}(Y = y)$ is satisfied when $\mathbb{P}(Y = y) = \mathbb{P}_{\mathcal{D}}(Y = y)$. That ends our proof. $\qquad\square$

## C   Useful Notations

We introduce some notations used in the proof of theorems.

**Notations** (Level sets).

$$\mathcal{S}_B(b) := \{x \in \mathcal{X} | \mathbb{P}_{\mathcal{D}}(Y = 0 | X^B = x^b) = b\}$$
$$\mathcal{S}_{f_B}(l) := \{x \in \mathcal{X} | p_0^b(X = x) = l\}.$$
$$\mathcal{S}_E(a) := \{x \in \mathcal{X} | \mathbb{P}_{\mathcal{D}}(Y = 0 | X = x) = a\}$$
$$\mathcal{S}_R(s) := \{x \in \mathcal{X} | \mathbb{P}(Y = 0 | X^S = x^s) = s\}$$

**Notation** ($\tilde{s}$).

$$\tilde{s}_{a,b} = \frac{a(1 - b)}{a(1 - b) + b(1 - a)}$$

**Notations** ($Y^S, \tilde{Y}, \hat{Y}$).

$$Y^S(x) := \operatorname{argmax}_{i \in \mathcal{Y}} \mathbb{P}_{\mathcal{D}}(Y = i | X^S = x^s), \tag{4}$$
$$\tilde{Y}(x) := \operatorname{argmax}_{i \in \mathcal{Y}} \mathbb{P}_{\mathcal{D}, f_M^*}(Y = i | X = x). \tag{5}$$
$$\hat{Y}(x) := \operatorname{argmax}_{i \in \mathcal{Y}} \mathbb{P}_{\mathcal{D}}(Y = i | X = x) \tag{6}$$

**Notation** ($\mathcal{P}^i(\cdot)$). Denote $\mathcal{P}^i(\mathcal{S}) := \mathbb{P}_{\mathcal{D}}(Y^S = i | \mathcal{S})$.

**Definition 1** (False Reversal Rate). *For an input $x$, we say $f_B(x)$ induces a false reversal if $\tilde{Y}(x) \neq \hat{Y}(x) = Y^S(x)$. The false reversal rate of a set $\mathcal{S}$ is defined as $\frac{\mathbb{P}_{\mathcal{D}}(\mathcal{S}_{fr})}{\mathbb{P}_{\mathcal{D}}(\mathcal{S})}$, where $x \in \mathcal{S}_{fr}$ if it occurs false reversal and $x \in \mathcal{S}$.*

Similarly we define the False Agreement Rate:

**Definition 2** (False Agreement Rate). *For an input $x$, we say $f_B(x)$ induces a false agreement, if $\tilde{Y}(x) = \hat{Y}(x) \neq Y^S(x)$. The false agreement rate of a set $\mathcal{S}$ is defined as $\frac{\mathbb{P}_{\mathcal{D}}(\mathcal{S}_{fa})}{\mathbb{P}_{\mathcal{D}}(\mathcal{S})}$, where $x \in \mathcal{S}_{fa}$ if it occurs false agreement and $x \in \mathcal{S}$.*

## D   Proof of Theorem 1

First we prove the following lemma.

**Lemma 1.** *Denote $\mathcal{R}_b(a) := \mathcal{P}^1(\mathcal{S}_E(a) \cap \mathcal{S}_B(b))$, and $p_B(a|b) := \mathbb{P}_{\mathcal{D}}(\mathcal{S}_E(a) | \mathcal{S}_B(b))$. We have*

$$\mathcal{R}_b(a) = \mathcal{P}^1(\mathcal{S}_R(\tilde{s}_{a,b})) = I(\tilde{s}_{a,b} < 0.5), \tag{7}$$
$$p_B(a|b) = C_{a,b} \mathbb{P}_{\mathcal{D}}(\mathcal{S}_R(\tilde{s}_{a,b})), \tag{8}$$

*where $C_{a,b} = \frac{1}{2}(\frac{a}{b} + \frac{1-a}{1-b})^{-1}$.*

*Proof.* For the first equation, it is obvious that $\mathcal{S}_E(a) \cap \mathcal{S}_B(b) = \mathcal{S}_R(\tilde{s}_{a,b}) \cap \mathcal{S}_B(b)$. Then we have

$$\mathcal{P}^1(\mathcal{S}_E(a) \cap \mathcal{S}_B(b)) = \mathcal{P}^1(\mathcal{S}_R(\tilde{s}_{a,b}) \cap \mathcal{S}_B(b)).$$

By the definition of $\mathcal{P}^1$ we have

$$\mathcal{P}^1(\mathcal{S}_R(\tilde{s}_{a,b}) \cap \mathcal{S}_B(b)) = \mathcal{P}^1(\mathcal{S}_R(\tilde{s}_{a,b})) = I(\tilde{s}_{a,b} < 0.5)$$

The first equation follows.

By $X^S \perp\!\!\!\perp X^B | Y$ on $\mathbb{P}_\mathcal{D}$, As $\mathbb{P}(Y = 0|X^S)$ is a function of $X^S$, $\mathbb{P}_\mathcal{D}(Y = 0|X^B)$ is a function of $X^B$, we have

$$\mathbb{P}_\mathcal{D}(Y = 0|X^S) \perp \mathbb{P}_\mathcal{D}(Y = 0|X^B)|Y \tag{9}$$

Without loss of generality, we can assume that $\mathbb{P}_\mathcal{D}(Y = 0|X^S)$ takes value in a discrete set $\mathcal{V}$. By the decomposition that

$$\mathbb{P}_\mathcal{D}(Y|X) \propto \mathbb{P}_\mathcal{D}(Y|X^B)\mathbb{P}_\mathcal{D}(Y|X^S)$$

We have

$$
\begin{aligned}
\mathbb{P}_\mathcal{D}(\mathcal{S}_E(a)|\mathcal{S}_B(b)) &= \sum_{i=0,1} \mathbb{P}_\mathcal{D}(\mathcal{S}_R(\tilde{s}_{a,b})|\mathcal{S}_B(b), Y = i)\mathbb{P}_\mathcal{D}(Y = i|\mathcal{S}_B(b)) \\
&= \sum_{i=0,1} \mathbb{P}_\mathcal{D}(\mathcal{S}_R(\tilde{s}_{a,b})|Y = i)\mathbb{P}_\mathcal{D}(Y = i|\mathcal{S}_B(b)) \\
&= \sum_{i=0,1} \frac{1}{2}\mathbb{P}_\mathcal{D}(Y = i|\mathcal{S}_R(\tilde{s}_{a,b}))\mathbb{P}_\mathcal{D}(\mathcal{S}_R(\tilde{s}_{a,b}))\mathbb{P}_\mathcal{D}(Y = i|\mathcal{S}_B(b)) \\
&= \frac{1}{2}(\tilde{s}_{a,b} \cdot b + (1 - \tilde{s}_{a,b})(1 - b))\mathbb{P}_\mathcal{D}(\mathcal{S}_R(\tilde{s}_{a,b})) \\
&= \frac{1}{2}(\frac{a}{b} + \frac{1-a}{1-b})^{-1}\mathbb{P}_\mathcal{D}(\mathcal{S}_R(\tilde{s}_{a,b})).
\end{aligned}
$$

The second equation follows. $\qquad\square$

Now we start the proof of the theorem.

*Proof.* Without the loss of generality, consider the case when $l_0 > 0.5$. The proof for $l_0 < 0.5$ has a symmetric form. Denote $\mathcal{R}(a) := \mathcal{P}^1(\mathcal{S}_E(a) \cap \mathcal{S}_{f_B}(l))$, and $p_f(a|l) := \mathbb{P}_\mathcal{D}(\mathcal{S}_E(a)|\mathcal{S}_{f_B}(l))$. The False Reversal Rate and False Agreement Rate on $\mathcal{S}_{f_B}(l)$ is

$$\mathcal{FR}(l) = \sum_a \mathcal{R}(a)p_f(a|l)I(a > l)I(a < 0.5) + (1 - \mathcal{R}(a))p_f(a|l)I(a < l)I(a > 0.5) \tag{10}$$

$$\mathcal{FA}(l) = \sum_a (1 - \mathcal{R}(a))p_f(a|l)I(a < l)I(a < 0.5) + \mathcal{R}(a)p_f(a|l)I(a > l)I(a > 0.5) \tag{11}$$

The total debiasing error on $\mathcal{S}_{f_B}(l)$ is the summation of False Reversal Rate and False Agreement Rate, which denoted as $E(l)$. The difference of total debiasing error at $l = a$ is

$$\Delta E(a) = (1 - 2\mathcal{R}(a))p_f(a|l) \tag{12}$$

$\Delta E(a) < 0$ when $\mathcal{R}(a) \in [0, 0.5), p_f(a|l) > 0$, $\Delta E(a) > 0$ when $\mathcal{R}(a) \in (0.5, 1], p_f(a|l) < 0$. By that, when $p_f(a|l) > 0, \forall a \in (0, 1)$, the total debiasing error is minimized at $a$ s.t. $\mathcal{R}(a) = 0.5$.

Denote $p_f^B(b|l) := \mathbb{P}_\mathcal{D}(\mathcal{S}_B(b)|\mathcal{S}_{f_B}(l))$. We have

$$\mathcal{R}(a) = \sum_b \mathcal{R}_b(a)p_B(a|b)p_f^B(b|l)\frac{1}{p_f(a|l)} \tag{13}$$

We suppose the support of $\mathbb{P}_\mathcal{D}(Y = 0|X^B)$ condition on $\mathcal{S}_{f_B}(l)$ is on $(l_0 - \epsilon, l_0 + \epsilon)$, i.e. $p_f^B(b|l)$ is non-zero only if $b \in (l_0 - \epsilon, l_0 + \epsilon)$.

By Lemma 1, we have

$$\mathcal{R}_b(a) = I(\tilde{s}_{a,b} < 0.5) \tag{14}$$

When $a \in (l_0 + \epsilon, 1)$, $\tilde{s}_{a,b} > 0.5, \forall b$. Thus $\mathcal{R}_b(a) = 1, \forall b$. We have

$$\mathcal{R}(a) = \sum_b p_B(a|b) p_f^B(b|l) \frac{1}{p_f(a|l)} = 1 \tag{15}$$

Similarly it can be derived that $\mathcal{R}(a) = 0$ when $a \in (0, l_0 - \epsilon)$. As a result, the debiasing error is non-decreasing as $l$ decreases on the interval $(0, l_0 - \epsilon)$ or increases on the interval $(l_0 + \epsilon, 1)$, i.e. The debiasing error increases as $|l - \mathbb{P}_{\mathcal{D}}(Y = 0|\mathcal{S}_{f_B}(l))|$ increases. Denote the absolute difference between $\mathbb{P}_{\mathcal{D}}(Y = 0|\mathcal{S}_{f_B}(l))$ and $l_{opt}$ which minimizes the debiasing error $E$ as $\delta(l_0, \epsilon, \alpha)$. As

$$\mathbb{P}_{\mathcal{D}}(Y = 0|\mathcal{S}_{f_B}(l)) = \sum_b b p_f^B(b|l) \in (l_0 - \epsilon, l_0 + \epsilon), \tag{16}$$

$l_{opt} \in (l_0 - \epsilon, l_0 + \epsilon)$, we have $\delta(l_0, \epsilon, \alpha) < 2\epsilon$.

Now we consider the case when $\alpha := \min_{X^S} \max_{i \in \{0,1\}} \mathbb{P}_{\mathcal{D}}(Y = i|X^S) > 0$, i.e. $\mathbb{P}_{\mathcal{D}}(\mathcal{S}_R(s)) = 0$ when $s \in (1 - \alpha, \alpha)$. When $\tilde{s}_{a,b} \in (1 - \alpha, \alpha)$, by Lemma 1 $p_B(a|b) = 0$, and we have

$$a \in \left( \frac{b}{\frac{1-b}{1-\alpha} + 2b - 1}, \frac{b}{\frac{1-b}{\alpha} + 2b - 1} \right) =: (L_{\alpha,b}, U_{\alpha,b}) \tag{17}$$

Both $L_{\alpha,b}$ and $U_{\alpha,b}$ increase as $b$ increase. As a result, for $\forall a \in (L_{\alpha,l_0+\epsilon}, U_{\alpha,l_0-\epsilon})$, $p_f(a|l) = \sum_b p_B(a|b) p_f^B(b|l) = 0$. When $l_0 - \epsilon = L_{\alpha,l_0+\epsilon}$, we have

$$\alpha = \frac{1}{2} + \frac{\epsilon}{2l_0(1 - l_0) + 2\epsilon^2} =: C_\alpha \tag{18}$$

When $l_0 + \epsilon = U_{\alpha,l_0-\epsilon}$, we also have $\alpha = C_\alpha$. As $L_{\alpha,l_0+\epsilon}$ decreases and $U_{\alpha,l_0-\epsilon}$ increases with $\alpha$, when $\alpha < C_\alpha$, we have the same conclusion: $\Delta E(a) \geq 0$ when $a \in (l_0 + \epsilon, 1)$ and $\Delta E(a) \leq 0$ when $a \in (0, l_0 - \epsilon)$, and $\delta(l_0, \epsilon, \alpha) < 2\epsilon$. That gives the first conclusion in Theorem 1.

For the case when $\alpha > C_\alpha$, $L_{\alpha,l_0+\epsilon} < l_0 - \epsilon$ and $U_{\alpha,l_0-\epsilon} > l_0 + \epsilon$. We consider the quantity $l_0 - \epsilon - L_{\alpha,l_0+\epsilon}$ and $U_{\alpha,l_0-\epsilon} - l_0 + \epsilon$. Denote $D(\alpha, l_0, \epsilon) = 2l_0 - (L_{\alpha,l_0+\epsilon} + U_{\alpha,l_0-\epsilon})$. We have

$$\frac{\partial D}{\partial \alpha} = \frac{(l_0 + \epsilon)(1 - l_0 - \epsilon)}{(1 - (l_0 + \epsilon) + 2(l_0 + \epsilon - 1)(1 - \alpha))^2} - \frac{(l_0 - \epsilon)(1 - l_0 + \epsilon)}{(1 - (l_0 - \epsilon) + 2(l_0 - \epsilon - 1)\alpha)^2} \tag{19}$$

$$= \frac{-(2l_0 - 1)[2\epsilon\alpha^2 + 2\alpha(l_0 + \epsilon)(l_0 - \epsilon) - 2\alpha(l_0 + \epsilon) - (l_0 + \epsilon)(l_0 - \epsilon) + l_0 + \epsilon]}{(1 - (l_0 + \epsilon) + 2(l_0 + \epsilon - 1)(1 - \alpha))^2(1 - (l_0 - \epsilon) + 2(l_0 - \epsilon - 1)\alpha)^2} \tag{20}$$

$$=: -\frac{1}{A}[2\epsilon\alpha^2 + 2\alpha(l_0 + \epsilon)(l_0 - \epsilon) - 2\alpha(l_0 + \epsilon) - (l_0 + \epsilon)(l_0 - \epsilon) + l_0 + \epsilon], A > 0 \tag{21}$$

There exists $\alpha'$ s.t. $\frac{\partial D}{\partial \alpha}(\alpha, l_0, \epsilon) < 0$ when $\alpha < \alpha'$, $\frac{\partial D}{\partial \alpha}(\alpha, l_0, \epsilon) > 0$ when $\alpha > \alpha'$. When $\alpha = C_\alpha$,

$$\frac{\partial D}{\partial \alpha}(C_\alpha, l_0, \epsilon) = \frac{2\epsilon(l_0 + \epsilon)(l_0 - \epsilon)(1 - (l_0 + \epsilon))(1 - (l_0 - \epsilon))}{A[2(l_0 + \epsilon)(l_0 - \epsilon) - 2l_0]^2} > 0 \tag{22}$$

Thus $D(\alpha, l_0, \epsilon) > D(C_\alpha, l_0, \epsilon) = 0$ when $\alpha > C_\alpha$. Denote $C := l_0 - \epsilon - \frac{l_0 + \epsilon}{(l_0 + \epsilon) + (1 - l_0 - \epsilon)\frac{\alpha}{1 - \alpha}}$, we have $l_{opt} \in (l_0 - \epsilon - C, U_{\alpha,l_0-\epsilon})$, as a result $C < \delta(l_0, \epsilon, \alpha) < 2\epsilon + C$. That ends our proof.

$\square$

# E    Proof of Theorem 2

*Proof.* As $\tilde{Y}(X) = 0$ if and only if $\mathbb{P}_{f_B}(Y = 0|X) > \mathbb{P}_{f_B}(Y = 1|X)$. The later is equivalent to

$$\mathbb{P}_{\mathcal{D}}(Y = 0|X)/q_0^b(x) > \mathbb{P}_{\mathcal{D}}(Y = 1|X)/q_1^b(x),$$

equivalently

$$\mathbb{P}_{\mathcal{D}}(Y = 0|X) > q_0^b(x).$$

As a result, when $\hat{Y}(X) \neq \tilde{Y}(X)$, we have

$$\mathbb{P}_{\mathcal{D}}(Y = \hat{Y}(x)|X = x) < q_{\hat{Y}(x)}^b(x)$$

Conversely, the above equation induces $\tilde{Y}(X) \neq \hat{Y}(X)$.

$\square$

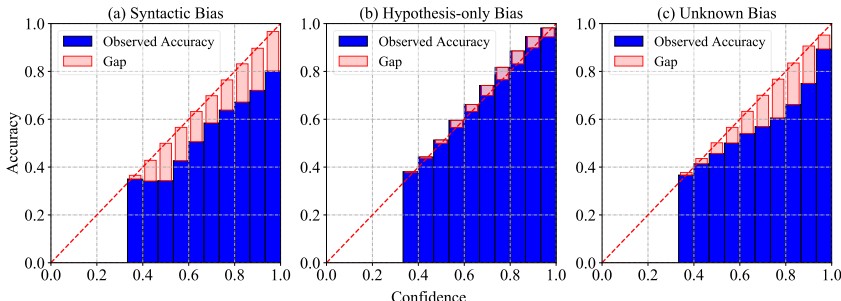

Figure 1: Reliability diagrams of the bias-only models on MNLI. On MNLI, (a) the syntactic bias-only model and (c) the unknown bias-only model are over-confident, (b) the hypothesis-only bias-only model is under-confident.

## F    Over- or Under-Confidence of the Bias-only Model on MNLI

We plot the confidence-reliability diagram [6] of these three models in Figure 1. The wide blue bars show the average accuracy of the bias-only model, and the narrow red bars show the gap between the average accuracy and the confidence of the bias-only model, i.e., the uncertainty estimation on the predicted class. For perfectly calibrated predictions, the curve in a reliability diagram should be as close as possible to the diagonal. Most of the blue bars below the diagonal indicate that the model is over-confident, otherwise is under-confident. It can be observed that the syntactic bias-only model and unknown bias-only model are over-confident, and the hypothesis-only bias-only model is under-confident.

## G    The Classification Accuracy of the Calibrated Bias-only Models

To facilitate the study, we demonstrate the classification accuracy of the calibrated bias-only models on different training datasets, as shown in Table 1. In the table, Un-Cal, Dirichlet, and TempS denote the bias-only model without calibration, with temperature scaling and Dirichlet calibrator, respectively.

Table 1: Accuracy of the calibrated bias-only models on different training datasets.

|          | FEVER | HANS | MNLI | Unknown |
|----------|-------|------|------|---------|
| Un-Cal   | 60.6  | 54.8 | 63.8 | 63.2    |
| TempS    | 60.6  | 54.8 | 63.8 | 63.2    |
| Dirichlet| 62.7  | 69.9 | 64.0 | 63.4    |

## H    Experiment on Image Classification

In image classification experiments, we validate the effectiveness of MoCaD on the texture bias in realistic images.

**Datasets**    We follow Bahng et al. [1] to conduct our experiment. The experiment is conducted on the 9-Class Imagenet dataset [1], which is a subset of ImageNet [5] containing 9 super-classes. The validation dataset and ImageNet-A [8] are used for evaluation. For the in-distribution validation dataset, an 'unbiased' accuracy measurement is used to evaluate the debiasing performance, denoted as Unbiased. It first obtains the proxy ground truths $c \in \{1, \ldots, K\}$ for texture bias using texture feature clustering. Then the dataset is grouped according to the texture-class combination $(c, y)$. The combination-wise accuracy $A_{c,y}$ is computed by $\mathrm{Corr}(c, y)/\mathrm{Pop}(c, y)$, where $\mathrm{Corr}(c, y)$ is the number of correctly predicted samples in $(c, y)$ and $\mathrm{Pop}(c, y)$ is the total number of samples in

$(c, y)$. Finally, `Unbiased` is the mean accuracy over all $A_{c,y}$ where the population $\text{Pop}(c, y) > 10$. Specifically, the texture features are extracted from images by computing the gram matrices of low-layer feature maps to capture the edge and color cues. It uses the feature maps from layer `relu1_2` of the ImageNet pre-trained `VGG16` [12]. The clustering process is done with the mini-batch k-means algorithm with $k = 9$ and batch size 1024. As k-means clustering is non-convex, the clustering is repeated three times with different initialization, and the averaged performance across the three trials is reported. ImageNet-A [8] is a dataset of natural adversarial filtered images that fool ImageNet-trained ResNet50 [7]. The images consist of many failure modes of networks when "frequently appearing background elements" [8] become erroneous cues for recognition.

**Main Model and Bias-only Model**    Following [1], the main model is a fully convolutional network followed by a global average pooling (GAP) layer and a linear classifier. Specifically, ResNet-50 architecture [7] is adopted as the main model. The bias-only model is a CNN with smaller receptive fields, which is expected to biased towards texture bias. Specifically, it is a `BagNet` [3], which is a variant of the ResNet50 architecture, by replacing many $3 \times 3$ with $1 \times 1$ convolutions, thereby limiting the receptive field size of the topmost convolutional layer.

Table 2: Classification accuracy on image classification.

| Method | ID | UnBiased | ImageNet-A |
|---|---|---|---|
| PoE | $94.6 \pm 0.2$ | $94.3 \pm 0.3$ | $31.8 \pm 1.9$ |
| **PoE**$_{\text{TempS}}$ | $94.7 \pm 0.3$ | $\mathbf{94.5} \pm 0.3$ | $\mathbf{31.9} \pm 1.1$ |
| **PoE**$_{\text{Dirichlet}}$ | $94.6 \pm 0.4$ | $94.3 \pm 0.4$ | $30.5 \pm 1.2$ |
| DRiFt | $94.6 \pm 0.2$ | $94.4 \pm 0.3$ | $31.9 \pm 0.8$ |
| **DRiFt**$_{\text{TempS}}$ | $94.8 \pm 0.4$ | $\mathbf{94.4} \pm 0.4$ | $\mathbf{32.5} \pm 1.2$ |
| **DRiFt**$_{\text{Dirichlet}}$ | $94.5 \pm 0.2$ | $94.3 \pm 0.2$ | $32.4 \pm 1.0$ |
| InvR | $94.5 \pm 0.4$ | $94.1 \pm 0.5$ | $31.6 \pm 0.3$ |
| **InvR**$_{\text{TempS}}$ | $94.3 \pm 0.1$ | $93.8 \pm 0.1$ | $\mathbf{32.2} \pm 1.5$ |
| **InvR**$_{\text{Dirichlet}}$ | $94.4 \pm 0.4$ | $\mathbf{94.2} \pm 0.2$ | $31.8 \pm 0.9$ |
| LMin | $90.9 \pm 0.5$ | $90.5 \pm 0.6$ | $27.7 \pm 1.6$ |
| **LMin**$_{\text{TempS}}$ | $91.1 \pm 0.6$ | $90.6 \pm 0.6$ | $\mathbf{28.1} \pm 1.8$ |
| **LMin**$_{\text{Dirichlet}}$ | $91.2 \pm 0.2$ | $\mathbf{90.9} \pm 0.2$ | $26.1 \pm 0.8$ |

**Training Details and Configurations**    We follow the configuration in [1]: the batch size is set to 128; learning rates are initially set to 0.001 and are decayed by cosine annealing and training for 120 epochs. As advised by Bahng et al. [1], we use AdamP optimizer [9] in the experiment. We experiment with 8 implementations of MoCaD, i.e. two different calibrators combined with four different ensembling strategies as the same as in previous experiments. For Learned-Mixin, the entropy term weight is set to the value suggested by [1]. We run each experiment five times and report the mean scores and the standard deviations. For the Dirichlet calibrator, we use the same configuration as in FEVER.

**Experimental Results**    Table 2 shows the experimental result on image classification. We can see that our MoCaD can achieve the best debiasing performance among all EBD methods, but the improvement is inconsistent. According to our theoretical analysis, that may because the invariant mechanism for image classification task has a higher certainty (bigger $\alpha$), reducing the impact of calibration error on debiasing.