# OpenReview forum: "Uncertainty Calibration for Ensemble-Based Debiasing Methods"
_NeurIPS.cc/2021/Conference — NeurIPS 2021 Poster_

### Official Review · Reviewer_CYB7 · 2021-06-28

**Rating:** 8
**Confidence:** 3

**Summary:**

This paper shows an increasing connection between the uncertainty estimation of the biased only models and proposed MoCaD, a three-stage EBD approach considering model calibrating.

**Limitations And Societal Impact:**

Yes.

**Main Review:**

Originality: High. The combination of uncertainty calibration and EBD is a very novel idea and the impressive results shown in this paper validates the effectiveness of such combination.

Quality: The submission is technically sound, with the major claims in Section 4 being backed up by both theoretical analysis and experimental results in Section 5. however, I do feel that the experiments section can be improved to reveal more insight into why MoCaD works well.

Clarity: This paper is well organized, although I struggled to remember the complicated notation system used in the paper. The authors should spend some time on improving the nomenclature even at the cost of abusing some notations, consider that various kinds of probabilities are needed in both main body and appendix. The paper does provide enough information to reproduce its results.

Significance: High. Debiasing is an increasingly important area in ML, and this work has the potential to serve as a foundational piece for future development.

Some areas where improvement can be done:

Experiments. From Theorem 1-2, the authors nicely connected the uncertainty calibration of bias-only models to the final performance of the debiased models, and motivated the use of calibration methods. In particular, in Section 6.2.1, the Dirichlet scaling method seems to work much better than the temperature scaling method. This is understandable, as Dirichlet scaling is more expressive, but it can also introduce some accuracy change that might hurt the debiasing model. Therefore, I would like to see  if more expressive calibration methods are indeed helpful, for example, the more expressive temperature scaling (http://proceedings.mlr.press/v119/zhang20k.html) and the nonparametric calibration <http://proceedings.mlr.press/v108/wenger20a.html>. Besides the class-wise ECE, the authors might also want to look into the confidence-based ECE (http://proceedings.mlr.press/v70/guo17a.html), which is related with theorem 2, as well as the accuracy of the calibrated bias-only model (to check if the calibration method introduces non-negligible hurt on that). Section 6.2.2 is indeed very informative and completely backs up Theorem 2, good job!

The title might be confusing. It might be interpreted to how to calibrate the uncertainty of final debiased model, while the uncertainty is in fact only calibrated for the bias-only model.

Some typos, e.g., Figure 3 'Syntatic Bias'.

------------After Rebuttal------------
I have read the authors rebuttal and appreciate the new experiments comparing accuracies. This moves the scores upward.

**Time Spent Reviewing:**

8

---

> ### Author Response · Authors · 2021-08-10
> **Response for Reviewer CYB7**
>
> We thank the reviewer for their time and positive comments. Please see below for responses to each point:
>
> **- I would like to see if more expressive calibration methods are indeed helpful, for example, the more expressive temperature scaling (http://proceedings.mlr.press/v119/zhang20k.html) and the nonparametric calibration http://proceedings.mlr.press/v108/wenger20a.html. Besides the class-wise ECE, the authors might also want to look into the confidence-based ECE (http://proceedings.mlr.press/v70/guo17a.html), which is related with theorem 2.**
>
> We thank the reviewer for this constructive comment. We would like to highlight related content in 6.1.1 where we show the effect of different extents of calibration (measured by the class-wise ECE). As our work inspires the combination of calibration and EBD methods, experimenting with more expressive calibration methods and more accurate calibration measurements could be an important future direction on the basis of this work, as suggested by the reviewer.
>
> **- the accuracy of the calibrated bias-only model (to check if the calibration method introduces non-negligible hurt on that).**
>
> We checked the accuracy of the calibrated bias-only model and report it in the following table. FEVER denotes the claim-only bias-only model on the FEVER training set. HANS, MNLI, and unknown denotes syntactic bias-only model, hypothesis-only model, and unknown bias-only model on MNLI training set correspondingly.
>
> |  | FEVER	| HANS | MNLI | Unknown |
> | -- | -- | -- | -- | -- |
> | Un-Cal | 60.6 | 54.8 | 63.8 | 63.2 |
> | Temps | 60.6 | 54.8 | 63.8 | 63.2 |
> | Dirichlet | 62.7 | 69.9 | 64.0 | 63.4 |
>
> The results show that the temperature scaling does not change the predicted label because the maximum of the softmax function remains unchanged. It only changes the uncertainty estimation. On the contrary, the Dirichlet calibrator will change the prediction accuracy. More precisely, the Dirichlet calibrator improves the accuracy of all bias-only models in our experiment.
>
> **- Some typos, e.g., Figure 3 'Syntatic Bias'.**
>
> We thank the reviewer for pointing this out. We will thoroughly check the paper and correct the typos.

---

### Official Review · Reviewer_P1n2 · 2021-07-07

**Rating:** 7
**Confidence:** 4

**Summary:**

This paper provides theoretical and empirical proof that the bias-only model used for two-step debiasing methods does not provide adequate and accurate uncertainty estimation. Via both mathematical proofs and empirical experimentation on two NLP tasks, the authors show that bias-only models have been over-estimated and overlooked, and that further work is needed to improve them.

**Limitations And Societal Impact:**

The fact that the paper focusses purely on NLP tasks and shows inconsistent improvement on image tasks is worrisome for me. What if the phenomena that were observed during the emprical experimentation were intrinsic to the nature of the two NLP tasks chosen (which is actually very few)? The fact that this method doesn't apply in image domains is worth exploring and explaining.

**Main Review:**

This paper provides a clear and interesting critique of the bias-only model used in two-step debiasing methods.

Some comments that I had while reading the paper:

Please note that  some recent works [22; 9] propose to jointly learn the bias-only model and the debiased main model  in an end-to-end manner. Since it is difficult to quantify the impact of the bias-only model in this scheme, we mainly focus on the two-stage methods  == How can your methodology be applied to these kinds of methods?

For Figure 1, wouldn't it be worth discussing the difference between the two plots (and the two tasks?)

Line 228-229: "In other words, it only changes the uncertainty estimation229 and *maintains* the model’s accuracy?
== not sure what 'remain' means

Why were fact verification and NLI the tasks chosen?

Is using a hand-crafted classifier for the NLI task comparable to a data-driven one for the fact verification one? Are they equivalent and comparable?

Any hypotheses why the method doesn't work as well on Learnd-Mixin on HANS? Is there a difference between that approach and others?

Line 359: "poorly calibrated"

"Further experimental results on image classification show inconsistent improvements" == maybe it depends on the image task? would it be worth trying tasks that are more complex and multi-modal, as opposed to simple classification? e.g. captioning or VQA?

What are some ways in which the bias-only model can be improved? I feel like there should be a lot more discussion of this, as well as the advantage of the joint bias and debiased model approach that was not studied at all in this paper.

**Time Spent Reviewing:**

1.5

---

> ### Author Response · Authors · 2021-08-10
> **Response for Reviewer P1n2**
>
> We thank the reviewer for their time and constructive comments. We respond below to each comment that was raised (reordered).
>
> **- Please note that some recent works [22; 9] propose to jointly learn the bias-only model and the debiased main model in an end-to-end manner. Since it is difficult to quantify the impact of the bias-only model in this scheme, we mainly focus on the two-stage methods == How can your methodology be applied to these kinds of methods?**
>
> Please note that calibration techniques for training ab-initio well-calibrated models [1] could be applied to the bias-only model in the end-to-end framework, instead of the post-hoc approach (Temperature Scaling and Dirichlet Calibration) used in this paper. However, a new theory needs to be developed to analyze the behavior of bias-only models in this case, as it involves learning dynamics. It could be an interesting new research direction, which is clearly beyond the scope of this paper.
>
> **- For Figure 1, wouldn't it be worth discussing the difference between the two plots (and the two tasks?)**
>
> The two plots here are just used to show the poor calibration phenomena, so we did not include a detailed discussion. Nevertheless, it is interesting to discuss the difference between the two plots. Remind that most of the blue bars below the diagonal indicate that the model is over-confident, otherwise is under-confident. Therefore, the left diagram shows the syntactic bias-only model is over-confident, while the other one is under-confident. The property of being over-confident or under-confident varies among different bias-only models on MNLI, as shown by Figure 1 in the Appendix. The reason why different models show different behavior remains a question, which is worth future investigation.
>
> **-Line 228-229: "In other words, it only changes the uncertainty estimation and maintains the model’s accuracy? == not sure what 'remain' means**
>
> We agree with the reviewer that “maintain” should be used here. Thanks for the correction.
>
> **- Why were fact verification and NLI the tasks chosen?**
>
> Fact verification and NLI are two common tasks used in debiasing, [2-4]. We follow these previous papers to choose these tasks and datasets for our experiments. We will add the explanations to the paper.
>
> **- What if the phenomena that were observed during the empirical experimentation were intrinsic to the nature of the two NLP tasks chosen (which is actually very few)?**
>
> We would like to illustrate the generality of our observed phenomena from the following two folds: 1) The existence of poorly calibrated machine learning models has been observed by much previous literature on calibration, as we mentioned in lines 181-184, which is independent of the task or dataset. 2) Our theoretical results are not restricted to any specific tasks or datasets. Though the latent constant $\alpha$ relates to the task, it only affects the numerical value of the threshold other than the conclusion of the theorem. In other words, calibration will benefit the debiasing performance for any task and bias-only models though the degree may vary. We would like to emphasize that the two tasks are chosen because they are representative tasks in debiasing. We have reported the result of 4 different bias-only models, 4 different EBD methods, and 2 different calibration methods, in total 32 different instances on the two representative tasks. These empirical results have shown consistent behavior of MoCaD on different configurations.
>
> **- Is using a hand-crafted classifier for the NLI task comparable to a data-driven one for the fact verification one? Are they equivalent and comparable?**
>
> The biases we considered on the two datasets are different. That is why different models are used for the bias-only model. In our opinion, it is not fair to compare the two bias-only models. Nevertheless, we would like to list their accuracies here. The hand-crafted classifier for the NLI task is designed to capture the syntactic bias, and the accuracy of it on the MNLI training set is 0.548. The claim-only classifier for the fact verification task is designed to capture the claim-only bias, and the accuracy of it on the FEVER training set is 0.606.
>
> **- Any hypotheses why the method doesn't work as well on Learned-Mixin on HANS? Is there a difference between that approach and others?**
>
> Yes, there is a difference. As we mentioned in line 127, in Learned-Mixin a trainable gate function is added as the exponent of the bias-only output. As a result, its optimal bias-only model is different from others and does not fit our theoretical assumptions. For a thorough comparison, we experimented with Learned-Mixin and reported the results. We will add these explanations to the result analysis part.
>
> **- Line 359: "poorly calibrated"**
>
> Sorry for the typos. Thanks for the correction.
>
> **- What are some ways in which the bias-only model can be improved? I feel like there should be a lot more discussion of this, as well as the advantage of the joint bias and debiased model approach that was not studied at all in this paper.**
>
> We would like to highlight that existing improvements (in a different aspect from this paper) of the bias-only model mainly focus on exploiting different prior knowledge to obtain bias-only models in the unknown bias case, as we introduced in Section 2, lines 79-85. We have experimented our method with the bias-only model proposed by Utama et al. [35] for the unknown bias case, and the result is shown in Table 1. It shows that our method consistently improves its out-of-distribution performance.
> For the joint bias and debiased model approach, we would like to highlight that it is an end-to-end debiasing framework, other than a method to improve the bias-only model. Currently, there is no conclusion to show the advantages of this framework with respect to the two-stage one or its effect on the bias-only model. For known dataset bias, [2-3] adopt the two-stage framework, while [5] adopts the end-to-end framework. For unknown dataset bias, [4,6] adopt the two-stage framework, while [7] adopts the end-to-end one. It could be an interesting new research direction to apply our methodology to the end-to-end methods, however as we illustrated in the answer to the first point, it is clearly beyond the scope of this paper.
>
> **- "Further experimental results on image classification show inconsistent improvements" == maybe it depends on the image task? would it be worth trying tasks that are more complex and multi-modal, as opposed to simple classification? e.g. captioning or VQA?**
>
> We agree with the reviewer, more complex and multi-modal tasks are worth trying in the future.
>
> **-The fact that this method doesn't apply in image domains is worth exploring and explaining.**
>
> We respectfully disagree that it has been shown as a fact that this method doesn't apply in image domains. The experimental result on the image classification task shows that our method can achieve the best performance among all EBD methods, though the improvement is not consistent across the calibrators. As we discussed in lines 367-371, a possible reason is that the invariant mechanism for image classification has a high certainty, reducing the impact of calibration error on debiasing according to our theoretical analysis.  As the reviewer suggested, more complex tasks are worth trying.
>
> [1] Kumar, A., Sarawagi, S., and Jain, U. Trainable calibration measures for neural networks from kernel mean embeddings. In International Conference on Machine Learning (ICML), pp. 2810–2819, 2018.
>
> [2] Christopher Clark, Mark Yatskar, and Luke Zettlemoyer. Don’t take the easy way out: Ensemble based methods for avoiding known dataset biases. In Proceedings of the 2019 Conference on Empirical Methods in Natural Language Processing and the 9th International Joint Conference on Natural Language Processing (EMNLP-IJCNLP), pages 4060–4073, 2019.
>
> [3] Prasetya Ajie Utama, Nafise Sadat Moosavi, and Iryna Gurevych. Mind the trade-off: Debiasing nlu models without degrading the in-distribution performance. arXiv preprint arXiv:2005.00315, 2020.
>
> [4] Prasetya Ajie Utama, Nafise Sadat Moosavi, and Iryna Gurevych. Towards debiasing nlu models from unknown biases. In Proceedings of the 2020 Conference on Empirical Methods in Natural Language Processing (EMNLP), pages 7597–7610, 2020.
>
> [5] Rabeeh Karimi Mahabadi, Yonatan Belinkov, and James Henderson. End-to-end bias mitigation by modelling biases in corpora. In Annual Meeting of the Association for Computational Linguistics, 2020.
>
> [6] Victor Sanh, Thomas Wolf, Yonatan Belinkov, and Alexander M Rush. Learning from others’ mistakes: Avoiding dataset biases without modeling them. In International Conference on Learning Representations, 2021.
>
> [7] Christopher Clark, Mark Yatskar, and Luke Zettlemoyer. Learning to model and ignore dataset bias with mixed capacity ensembles. In Proceedings of the 2020 Conference on Empirical Methods in Natural Language Processing: Findings, pages 3031–3045, 2020.

---

### Official Review · Reviewer_7E1r · 2021-07-14

**Rating:** 6
**Confidence:** 3

**Summary:**

The paper reveals that uncertainty calibration of the biased model is important for debiasing performance in EBD methods. By calibrating the biased model, the proposed method improves the performance of existing methods.

**Limitations And Societal Impact:**

Yes, the authors have addressed the limitations of the paper especially the inconsistent results on image data.

**Main Review:**

Originality: to the best of the reviewer's knowledge, this paper is the first to discuss calibration in EBD methods even though the topic of calibration has been studied extensively.  To some extent, the paper lacks novelty because the paper uses existing techniques without modification to improve existing EBD models.

Clarity: the paper is well organized and easy to follow. The method starts with the formalization of the EBD (Sec.3) method and theoretical (Sec.4.1) and empirical analysis (Sec.4.2) of the bias-only model. Then it introduces calibration techniques and the complete pipeline (Sec.5). However, there are some technical details that require further clarification. Please see the questions.

Quality: the paper conducts theoretical analysis and verifies assumptions empirically.

Significance: the paper points out an overlooked problem in EBD methods. The work can inspire further investigation of calibration problems in ensemble-based debiasing methods.

Questions:

1, Is the predicted label $Y(s)$ from the intrinsic model (line 136) different from the label $\tilde{Y}(x)$ predicted by the ideal predictor (line 166)? Should they be the same?

2, The global certainty level $\alpha:=min_{X^s} max_{i\in\{0,1\}} P_D(Y=i|X^s)$ (line 149) is the least confidence prediction from the true principle.  Can the author justify its role as a "global" certainty level? It's not clear why this quantity conveys a "global" view of certainty.

2, Regarding Figure 3: the plot shows monotonically increasing performance with temperature. As the temperature increases,  the biased models will become more and more uncertain. At some point, this must hurt the debiasing performance because when the distribution of the biased model is almost flat, it does not provide any information to the unbiased model.  Can the authors comment on this thought?

**--------------------post-rebuttal review--------------------**

I appreciate the authors' response and discussion on my questions. I think the paper uncovers an overlooked problem in EBD methods and can inspire further research.

**Time Spent Reviewing:**

1.5 hour

---

> ### Author Response · Authors · 2021-08-10
> **Response for Reviewer 7E1r**
>
> We thank the reviewer for the insightful comments. We are pleased to note the positive remarks on the significance of our work. Please see below for a detailed response to each point that was raised:
>
> **- Is the predicted label from the intrinsic model (line 136) different from the label predicted by the ideal predictor (line 166)? Should they be the same?**
>
> They are different. Intuitively, the intrinsic model is unbiased and with invariant out-of-distribution performance, while the ideal predictor is biased to some features. Their relationship is shown in line 100.
>
> **- The global certainty level (line 149) is the least confidence prediction from the true principle. Can the author justify its role as a "global" certainty level? It's not clear why this quantity conveys a "global" view of certainty.**
>
> Here $\alpha$ is taken as the “global” minimum of $\max_{i \in \{0,1\}} P_D(Y = i|X^S)$ on the whole space of $x^s$, so the word “global” is used for underlining this property.
>
> **- Regarding Figure 3: the plot shows monotonically increasing performance with temperature. As the temperature increases, the biased models will become more and more uncertain. At some point, this must hurt the debiasing performance because when the distribution of the biased model is almost flat, it does not provide any information to the unbiased model. Can the authors comment on this thought?**
>
> We thank the reviewer for this insightful thought. It is true that when the temperature in the temperature scaling is extremely high, the distribution of the bias-only model (biased model) is almost flat, and the debiasing performance will be hurt. We would like to highlight that the Temperature Scaling algorithm generally converges to a finite temperature that minimize the calibration error [1]. For example in our experiments, the optimal temperatures given by the Temperature Scaling algorithm are 0.915, 1.585, 1.646, and 0.792, for the hypothesis-only model, syntactic bias-only model, unknown biased model on MNLI, and the claim-only model on FEVER, respectively.
>
> [1] Chuan Guo, Geoff Pleiss, Yu Sun, and Kilian Q. Weinberger. On calibration of modern neural networks. In Proceedings of the 34th International Conference on Machine Learning (ICML'17). 2017.

---

### Official Review · Reviewer_x4rW · 2021-07-21

**Rating:** 6
**Confidence:** 3

**Summary:**

- The paper analyzes ensemble-based debiasing (EBD) methods for training models that don’t rely on dataset biases and thus has good out-of-distribution performance
- The theoretical analysis shows that the debiased model performance decreases monotonically as calibration error increases, and properly calibrating the model can improve not just out-of-distribution performance, but also in-distribution performance under certain conditions
- Based on the theoretical analysis, the authors propose a modification to EBD methods, where the bias-only model is calibrated. Empirically, this improves the performance of debiased models.


**Limitations And Societal Impact:**

Yes

**Main Review:**

- The paper highlights the importance of calibrating bias-only models for EBD methods. Although this is not so surprising given that EBD methods rely on probabilistic estimates outputted by bias-only models, the findings (both theoretical and empirical) are novel to my knowledge.
- The theoretical analysis clearly illustrates the benefits of calibrating the bias-only model both for out-of-distribution and in-distribution model, and this directly motivates the proposed modification to the EBD methods. The modification (adding a calibration step) is simple, but well-motivated, practical, and novel to my knowledge.
- The paper includes thorough, systematic experiments showing that the proposed modification improves the debiased model performance, in line with the theoretical analysis. The empirical improvement in performance is small, especially relative to the empirical gains yielded by the standard EBD methods, and this could make the key takeaway of the paper limited in its impact. However, the empirical gains are consistent across multiple datasets, calibration methods, and debiasing methods. They also have additional empirical analysis, which are also consistent with the theoretical results.
- The paper is clearly written, and I enjoyed reading it!

Minor comments
- For Theorem 1, what kind of values should I expect for the threshold? I understand it depends on latent constants, but it’d be nice to get a rough range.
- It took me some time to parse some of the theoretical notation and statements. For example, I found the notation for debiasing performance in Theorem 1 unclear.


**Time Spent Reviewing:**

2

---

> ### Author Response · Authors · 2021-08-10
> **Response for Reviewer x4rW**
>
> We appreciate the positive feedback and we are happy the reviewer has highlighted the clarity of the paper, the novelty of our findings and method, as well as the thoroughness of our experiments. Below, we respond to each comment that was raised.
>
> **- For Theorem 1, what kind of values should I expect for the threshold? I understand it depends on latent constants, but it’d be nice to get a rough range.**
>
> We agree with the reviewer that a rough range can benefit. However, as the latent constants are related to the unknown posterior characteristics of the bias-only model, we can only empirically show some statistics of the threshold. Among the three latent constants $\alpha, l_0,$ and $\epsilon$, $\alpha$ can not be observed even empirically since the intrinsic principle $P_D(Y|X^S)$ is unavailable. As a result, we calculate the empirical values of $l_0, \epsilon$ of the syntactic bias-only model on MNLI, and report the value of $2\epsilon$ as an approximation of the threshold. Specifically, we use the Mean-shift algorithm to cluster the instances represented by hand-crafted features to discretization $X^B$, and make use of the data binning algorithm in [1] to create $S_{f_B}(l)$. The result shows that the weighted average value of $2\epsilon$ is 0.064, the maximum is 0.51, the minimum is 0, and the weighted median is 0.012.
> The following table shows the statistics of the 5 biggest sets.
>
> | Ratio   |   $l_0$    |     $2\epsilon$	|    >$2\epsilon?$ |
> | -- | -- | -- | -- |
> | 0.092 | 0.186	| 0.024 | true |
> | 0.092 | 0.256	| 0.079 | true |
> | 0.090 | 0.267	| 0.012 | true |
> | 0.085 | 0.377 | 0.126 | true |
> | 0.084 | 0.143	| 0.005 | true |
>
> Here the first column shows the relative size of the set with respect to the whole dataset. The last column shows whether $|l - P_D(Y=0|S_{f_B}(l))| > 2\epsilon$ holds on the set.  The result shows this equation holds on sets that contain 83.4% MNLI samples (327352) in total. In conclusion, these empirical statistics suggest that the threshold can be relatively small in most cases so that the calibration can improve the debiasing performance.
>
> **- It took me some time to parse some of the theoretical notation and statements. For example, I found the notation for debiasing performance in Theorem 1 unclear.**
>
> We thank the reviewer for raising this point. In Theorem 1 we denote the debiasing performance on $S_{f_B}(l)$ as $P_D(\{x \in S_{f_B}(l)|\tilde{Y}(x) = Y(x)\})$, where {$x \in S_{f_B}(l)|\tilde{Y}(x) = Y(x)$} denotes the subset of $S_{f_B}(l)$ on which the main model gives the same prediction as the unbiased model. The debiasing performance on $S_{f_B}(l)$ is then defined as the probability of this set. We will add the explanation to the paper for better readability here.
>
> [1] Chuan Guo, Geoff Pleiss, Yu Sun, and Kilian Q. Weinberger. On calibration of modern neural networks. In Proceedings of the 34th International Conference on Machine Learning (ICML'17). 2017.

---

### Decision · Program_Chairs · 2021-09-27

**Decision:**

Accept (Poster)

**Comment:**

This paper studies the uncertainty calibration properties of the bias-only model in ensemble-based de-biasing methods. The authors argue that the accurate estimation of uncertainty in the bias-only model has been overlooked in prior research, but can have an important influence on the overall de-biasing process. The method proposed in the paper applies calibration methods to the output of the bias-only model. Thus, from a novelty perspective, the method itself is a novel combination of existing approaches, while at a conceptual level the paper is addressing a previously overlooked issue and thus has stronger novelty at this level. The reviewers judged the approach and the theoretical development to be technically correct. The experiments were judged to provide adequate support for claims. The writing was judged to be clear. The reviewers indicated that all of their questions had been adequately addressed following the author response. As a result, the paper is recommended for acceptance. The authors should be sure to incorporate the discussed clarifications and updates in the final manuscript.